# Pathophysiology of Cerebellar Degeneration in Mitochondrial Disorders: Insights from the *Harlequin* Mouse

**DOI:** 10.3390/ijms241310973

**Published:** 2023-06-30

**Authors:** Miguel Fernández de la Torre, Carmen Fiuza-Luces, Sara Laine-Menéndez, Aitor Delmiro, Joaquín Arenas, Miguel Ángel Martín, Alejandro Lucia, María Morán

**Affiliations:** 1Mitochondrial and Neuromuscular Diseases Laboratory, Instituto de Investigación Sanitaria Hospital ‘12 de Octubre’ (‘imas12’), 28041 Madrid, Spain; mftorre@ing.uc3m.es (M.F.d.l.T.); cfiuza.imas12@h12o.es (C.F.-L.); sara.laine@cbm.csic.es (S.L.-M.); adelmiro@h12o.es (A.D.); joaquin.arenas@salud.madrid.org (J.A.); mamcasanueva.imas12@h12o.es (M.Á.M.); 2Spanish Network for Biomedical Research in Rare Diseases (CIBERER), U723, 28029 Madrid, Spain; 3Servicio de Bioquímica Clínica, Hospital Universitario “12 de Octubre”, 28041 Madrid, Spain; 4Servicio de Genética, Hospital Universitario “12 de Octubre”, 28041 Madrid, Spain; 5Faculty of Sports Sciences, European University of Madrid, 28670 Madrid, Spain; alejandro.lucia@universidadeuropea.es; 6Spanish Network for Biomedical Research in Fragility and Healthy Aging (CIBERFES), 28029 Madrid, Spain

**Keywords:** mitochondrial diseases, OXPHOS disorders, complex I, *Harlequin* mouse, ataxia, long-term depression, glutamate, GABA

## Abstract

By means of a proteomic approach, we assessed the pathways involved in cerebellar neurodegeneration in a mouse model *(Harlequin*, *Hq*) of mitochondrial disorder. A differential proteomic profile study (iTRAQ) was performed in cerebellum homogenates of male *Hq* and wild-type (WT) mice 8 weeks after the onset of clear symptoms of ataxia in the *Hq* mice (aged 5.2 ± 0.2 and 5.3 ± 0.1 months for WT and *Hq*, respectively), followed by a biochemical validation of the most relevant changes. Additional groups of 2-, 3- and 6-month-old WT and *Hq* mice were analyzed to assess the disease progression on the proteins altered in the proteomic study. The proteomic analysis showed that beyond the expected deregulation of oxidative phosphorylation, the cerebellum of *Hq* mice showed a marked astroglial activation together with alterations in Ca^2+^ homeostasis and neurotransmission, with an up- and downregulation of GABAergic and glutamatergic neurotransmission, respectively, and the downregulation of cerebellar “long-term depression”, a synaptic plasticity phenomenon that is a major player in the error-driven learning that occurs in the cerebellar cortex. Our study provides novel insights into the mechanisms associated with cerebellar degeneration in the *Hq* mouse model, including a complex deregulation of neuroinflammation, oxidative phosphorylation and glutamate, GABA and amino acids’ metabolism

## 1. Introduction

Mitochondrial diseases (MD) are a heterogeneous group of disorders with genetic origin associated with a failure in the oxidative phosphorylation (OXPHOS) system function [1,2]. Although rare in the general population, MD are the most common error in congenital metabolism. Among them, complex I deficiency accounts for ~30% of MD’s diagnosed cases, and it is considered the most frequent cause of these disorders [3,4,5]. The tissues most affected by mitochondrial dysfunction are those with higher oxygen and energy demands, such as nervous tissue, muscle, heart and liver tissues [1,6,7]. Particularly, the nervous system energy requirements are higher than any other tissue or organ [8,9,10], and neuronal function relies on OXPHOS activity for processes such as ion transport, synaptic trafficking, vesicle release and neurotransmitter recycling [11,12,13,14,15,16]. In addition, the OXPHOS system failure frequently also leads to excessive reactive oxygen species (ROS) production, which, due to the lower efficiency of neuronal antioxidant detoxification systems, makes them more susceptible than other cells to oxidative stress and contributes to modifying normal neuronal activity, leading to neurodegeneration [17]. Moreover, a slight decrease in complex I activity has been reported to be enough to alter neuronal homeostasis, increase glutamate release and trigger excitotoxicity [3,18,19].

The *Harlequin* (*Hq*) mouse is a model of MD that harbors a proviral insertion in the *Aifm1* gene, which leads to a ~80% decrease in AIF protein levels in distinct tissues [20,21,22]. In physiological conditions, AIF is anchored in the inner mitochondrial membrane, and it is essential for the optimal function of the OXPHOS system, presumably through its interaction with mitochondrial import protein MIA40 [23,24,25,26,27,28]. As a result of AIF deficiency, the *Hq* mouse presents a deficit in complex I levels and activity in several tissues and organs, which in the cerebellum also leads to neuroinflammation and oxidative stress [20,22,29]. These metabolic disturbances trigger a neurodegenerative process in the *Hq* mouse that starts with granular cell death and progresses with Purkinje cell degeneration leading to cerebellar atrophy and ataxia [20,30].

The aim of the present study was to assess by means of a proteomic approach the cellular pathways implicated in the neuronal death secondary to the complex I deficiency in the *Hq* mouse model of MD in order to identify potential therapeutic targets to attenuate neurodegeneration. The results obtained allow us to identify previously unreported disturbances in proteins involved in glutamate and GABA metabolism and neurotransmission, as well as changes in most proteins involved in the cerebellar mechanism of synaptic plasticity known as long-term depression (LTD), which are relevant for motor learning and are also altered in other models of ataxia.

## 2. Results

### 2.1. Proteomic Analysis

In order to unveil potential disturbances induced by AIF deficiency in cellular pathways, the differential proteomic profile of the cerebellum was semiquantitatively analyzed by iTRAQ technology. The main differences detected between *Hq* animals and their WT littermates are shown in Table 1. The proteins showing the highest relative differences between *Hq* and WT belonged to the following main categories: glial activation; neurotransmission; cellular signaling and Ca^2+^ handling; osmotic regulation; OXPHOS system; and miscellaneous. To validate the main results of the proteomic analyses, we selected some candidates from each category for further biochemical study. Some proteins that showed a low proportion of peptides in the *Hq* group in comparison to the WT group in the identification prior to the iTRAQ labeling were also validated (EAAT2, mGluR1 and GluRδ2).

#### 2.1.1. Alterations in Proteins Involved in Glutamate and Gamma Aminobutyric Acid Neurotransmission in the Cerebellum

Several proteins related to glutamate and gamma aminobutyric acid (GABA) neurotransmission were differentially expressed due to AIF deficiency, and the changes in their levels were validated in subsequent Western blotting analyses.

Regarding glutamatergic neurotransmission, we found significantly lower levels of both excitatory amino acid transporters, EAAT2 (also known as Slc1a2) and EAAT4 (also known as Slc1a6) (*p* < 0.0001 for both variables) in the *Hq* cerebellum of 5-month-old animals compared to their WT littermates (Figure 1A). In the study of the evolution of neurodegeneration, similar changes were present in EAAT4 but not in EAAT2 from 2 months of age (Table 2), but *Slc1a2* gene expression was significantly higher at 6 months but not in 2- and 3-month-old *Hq* animals (relative expression 100 ± 6 and 150 ± 2.8 in WT and *Hq,* respectively; *p* < 0.004 Mann–Whitney U test). The EAAT4’s lower protein content in the *Hq* cerebellum was further confirmed by a qualitative assessment of tissue sections, in which EAAT4 labeling showed a restricted expression in Purkinje cell dendrites and soma, and a lower signal intensity in the *Hq* cerebellum (Figure 1B).

On the other hand, we observed higher levels of GABA transporter 3 (GAT-3, also known as sodium- and chloride-dependent GABA transporter [Slc6a11]) (*p* < 0.0001) and of vesicular GABA transporter (VGAT, also known as Slc32a1) (*p* = 0.0021) at 5 months in the *Hq* cerebellum compared to WT tissue (Figure 2). The study of the time course of the disease corroborated these results showing similar disturbances in both variables at 6 months but no changes at earlier stages (Table 2).

Overall, the present biochemical analyses revealed noteworthy variations in proteins involved in glutamate and GABA neurotransmission in the *Hq* mouse model. The earliest change observed was the lower level of EAAT4 transporter at 2 months of age, while lower levels of EAAT2, GAT-3 and VGAT were only present at 5–6 months of age.

#### 2.1.2. Disturbances in Proteins Involved in Synaptic Plasticity and Neuronal Ca^2+^ Homeostasis in the Cerebellum

The proteomic analysis also identified a differential expression of a set of proteins involved in intracellular Ca^2+^ handling and synaptic plasticity in 5-month-old *Hq* vs. WT mice: the metabotropic glutamate receptor 1 (mGluR1), the glutamate ionotropic receptor delta type subunit 2 (GluRδ2), Homer protein homolog 3 (HOMER-3) and the inositol 1,4,5-trisphosphate receptor type 1 (IP3R1) (Table 1). Immunodetection by Western blotting demonstrated a significant group effect for all the proteins analyzed (*p* < 0.0001 for all variables; except for IP3R1, with *p* = 0.0472), with significantly lower levels in the *Hq* cerebellum compared to WT littermates (Figure 3). We also analyzed the levels of two proteins that are substrates of the active form of PKC, the myristoylated alanine-rich C-kinase substrate phosphorylated at serine 152/156 (pMARCKS) and the subunit 2 of the glutamate ionotropic receptor AMPA type phosphorylated at serine 880 (pGRIA2). The results obtained showed significantly lower levels of both phosphorylated proteins (*p* < 0.0001 for both variables) in the *Hq* cerebellum than in WT mice (Figure 3), which corroborated a lower activity of PKC in the *Hq* cerebellum. We previously described in these animals normal levels of calbindin but a lower number of Purkinje cells at this age [29]. To assess if the changes observed in the Purkinje cells’ proteins involved in neurotransmission and Ca^2+^ homeostasis could be caused by a degeneration of these neurons, we normalized its levels by calbindin as a Purkinje cell marker and by the number of calbindin positive cells per mm measured in sagittal sections of the cerebellum (Appendix A). The normalized values also revealed significant differences for all variables except for IP3R, which almost reached significance, supporting that the detected changes in these proteins were not attributable to Purkinje cell loss. In the time-course study of the neurodegeneration, we observed similar changes in all these variables at 6 months of age, but, surprisingly, only EAAT4 and pGRIA2 were lower in the *Hq* cerebellum in comparison with the WT cerebellum at 2 and 3 months of age, despite the absence of neuron loss at these early stages, according to the granular and Purkinje cells markers NeuN and Calbindin (Table 2). Finally, the expression of the Grm1 gene encoding mGluR1 was also analyzed at 2, 3 and 6 months of age, showing only a Grm1 downregulation in 6-month-old *Hq* mice (relative gene expression 100 ± 14 and 60 ± 9.5 in WT and *Hq*, respectively; *p* < 0.03 Mann–Whitney U test).

#### 2.1.3. Disturbances in Other Cerebellar Proteins

Finally, we also aimed to validate the observed changes in the proteomic analysis for some proteins of the remaining categories, osmotic regulation (serum albumin, aquaporin-4 [AQP4] and miscellaneous group (β-globin, α-globin 1, apolipoprotein A-I and β-globin). However, only AQP4 immunodetection revealed a significant group effect (*p* < 0.0001), with higher levels in *Hq* than in WT mice (Figure 4). 

### 2.2. Amino Acid and Amino Acid Derived Metabolites Content in the Cerebellum and Brain

To have insight in the effects of the mitochondrial defect on the cerebellar and brain metabolism, the levels of amino acids and amino acid derived metabolites were measured in tissue homogenates of 6-month-old mice. At that age, we observed higher levels of ammonium and GABA (*p* = 0.0079 and *p* = 0.0317, respectively) in the *Hq* cerebellum in comparison to WT tissue (Table 3). Conversely, glutamate and serine levels (*p* = 0.0079, both variables) were significantly lower in the *Hq* mice’s cerebellum, while the levels of the remaining of metabolites were unaltered (Table 3). On the contrary, in the 6-month-old *Hq* brain, we found only higher levels of alanine and glutamine (*p* = 0.0079 and *p* = 0.0317, respectively), compared to those of the WT brain, and an absence of differences in the other metabolites (Table 3). Data obtained in the 6-month-old *Hq* brain were similar to the results observed in the cerebellum at 2 months of age, where we detected higher levels of alanine, glutamine and threonine (*p* = 0.0286, *p* = 0.0079 and *p* = 0.0079, respectively) and no changes in other metabolites analyzed (Table 3). To assess if granular cell loss at 6 months could have influenced the glutamate content in the *Hq* cerebellum, we normalized glutamate by NeuN. The results showed a higher glutamate content in the *Hq* group, but this difference did not reach statistical significance (Appendix A).

## 3. Discussion

The goal of the present study was to provide novel insights into the pathophysiology of cerebellar alterations induced by AIF deficiency in the *Hq* mouse model of MD by means of a proteomic approach, which could help to identify new therapeutic targets to prevent neuronal death. In our work, we found unreported disturbances in cerebellar metabolism and proteins involved in neurotransmission, in Ca^2+^ homeostasis, in the LTD mechanism and in osmotic regulation.

Since its first description by Klein et al., it has been known that oxidative stress is a key factor in the neurogenerative process of the *Hq* mouse model, in which AIF deficiency leads to an increased peroxide sensitivity of granular cells, higher levels and activity of antioxidant enzymes and a progressive increase in ROS-induced damage in neurons [20]. The involvement of oxidative stress in this model of MD was later corroborated by other groups, not only in the cerebellum but also in the retina and in other tissues [22,29,31,32,33,34]. In the current study, the main changes observed in the iTRAQ-identified proteins did not include any oxidative-stress-related protein. The first alteration observed in a protein related to oxidative stress was detected for peroxiredoxin 6, with an *Hq*/WT ratio around 1.25 (already described by our group [29]), followed by gluthathione-S transferase P with an *Hq*/WT ratio around 1.12 and by smaller changes in other proteins. Since the iTRAQ technique does not identify all the proteins present in the tissue, we cannot rule out significant alterations in the levels of undetected antioxidant proteins in the present study. On the other hand, it is possible that among the proteins identified by iTRAQ in which the relative amount of protein did not change between both experimental groups, some protein could experience changes in their oxidation state. Therefore, carrying out studies to analyze post-translational oxidative modifications in the cerebellar proteome of the *Hq* mouse could help to shed light on the involvement of oxidative stress in the neurodegenerative process of this MD model.

An important finding in our study was the lower content of glutamate transporters EAAT2 and EAAT4 in the *Hq* cerebellum that could suggest an altered glutamate uptake. Functional disturbances in EAAT-dependent glutamate uptake was previously described in cybrids harboring Leber hereditary optic neuropathy mutations, in which complex-I-deficiency-induced oxidative stress led to an EAAT1 inhibition and a lower glutamate uptake capacity, a phenomenon that was proposed to underlie neuron death [35,36]. In contrast to our results, these studies reported no alterations in the protein levels of glutamate transporters, but it is possible that chronic oxidative stress in the *Hq* cerebellum might induce, in the long term, a degradation of these transporters. Moreover, astroglial activation, a phenomenon that occurs in the *Hq* mouse [29,30,37], has been reported to induce EAAT2 and EAAT4 downregulation in the cerebellum [38]. The observed lower EAAT4 levels at 2 months in the *Hq* cerebellum suggests that it might be more sensitive than EAAT2 to oxidative stress and might result in an early, low glutamate uptake capacity that could promote excitotoxicity and cell death at later stages. Of note, neuronal death mediated by the loss of EAAT2 and excitotoxicity has been previously described in several neurodegenerative diseases [39,40,41]. Moreover, complex I deficiency can increase the release of glutamate in the synaptic terminal and would result in a loop of excitotoxicity [3,18,19,42]. In this regard, MD patients with mitochondrial encephalomyopathy, lactic acidosis and strokelike episodes (MELAS) syndrome show higher glutamate levels in cerebrospinal fluid, demonstrating increased glutamate release in the nervous system of some MD [43]. Future studies assessing whether excitotoxicity occurs in glutamatergic synapses in the cerebellum of the *Hq* mouse are needed.

Despite the lower content of EAAT transporters in the *Hq* cerebellum, glutamate levels were lower compared to those of WT mice. Similar results have been observed in other animal models of cerebellar ataxia [44,45], in the hippocampus of patients with Alzheimer’s disease [46] and in the brain of Ndufs4−/− [47] and mutator mice models of MD [48]. In the present work, glutamate depletion could derive from a failure in its biosynthesis, but the normal levels of its precursor, glutamine, in the *Hq* cerebellum argues against this possibility. Another possible explanation for glutamate loss in the *Hq* mouse could be the death of more than 50% of the granular cell population in 6-month-old mice [29], which is the main source of glutamate in cerebellar cortex. In the present work, glutamate content normalized by the granular cell marker NeuN showed a higher glutamate content in the *Hq* group, although this difference did not reach significance, suggesting a role of excitotoxicity in the neurodegenerative process of the *Hq* mouse. To the altered glutamate levels could contribute an enhanced conversion rate of this amino acid into the Krebs cycle intermediate α-ketoglutarate to increase ATP synthesis [49]. In this regard, a mathematical simulation of complex-I-deficient human cardiomyocytes has shown that amino acid supplementation, through its conversion into glutamate, can enhance ATP synthesis [50], and in the muscle from patients with severe mitochondrial myopathy, a higher glutamate content for energy generation and NADH oxidation has been recently described [51]. Moreover, a Krebs cycle anaplerosis with α-ketoglutarate in cybrids with a mutation in the MT-ATP6 gene and in a murine model of COX10−/− mitochondrial myopathy has demonstrated the ability to attenuate the alterations derived from mitochondrial failure [52]. Moreover, in MELAS patients, low glutamine is detected in cerebrospinal fluid and can be normalized with its oral supplementation, findings attributed to an increased consumption of this amino acid in the central nervous tissue of these patients to act as glutamate precursor [43,53].

The enhanced glutamine deamination into glutamate could also explain the higher ammonium content observed in the *Hq* cerebellum. It is relevant to highlight that ammonium is neurotoxic at high concentrations [54,55,56], chronically triggers neuroinflammation and enhances proinflammatory cytokine release [56,57], conditions that occur in the *Hq* cerebellum [29]. The higher levels of GABA and its transporters observed at 6 months in the *Hq* cerebellum in the present work could also be related to hyperammonemia, which has been previously reported to induce disturbances in glutamate and GABA transporters and to indirectly contribute to coordination failure and cerebellar ataxia [56,57,58,59]. In addition, the higher GABA levels in the *Hq* cerebellum could also by explained by an enhanced neurotransmitter synthesis by the glutamate decarboxylase enzyme, as an attempt to increase succinate production to feed complex II, bypassing complex I and promoting ATP synthesis [60,61,62]. Of note, novel alterations in three key clusters of genes and cellular functions encompassing glutamate, glutamine, GABA metabolism and the Krebs cycle have been recently described in MELAS cybrids, changes that can be partially mitigated by a Krebs cycle anaplerosis with ketones [63]. These results support that in some MD, the mitochondrial defect leads to a Krebs cycle blockade and compensatory mechanisms involving glutamate and GABA. In the present study, the low serine content observed in the cerebellum of 6-month-old *Hq* mice also points to an abnormal amino acid metabolism and correlates with previous findings of disturbed folate-driven 1C metabolism involving a higher serine utilization in AIF-deficient mice [64]. Moreover, the higher alanine and glutamine content in the cerebellum of 2-month-old and the brain of 6-month-old *Hq* mice could suggest that in the cerebellum, in early stages of the disease and in less affected areas of the central nervous system, there is an accelerated glutamine anaplerosis as a compensatory mechanism leading to a higher alanine production, as previously described in MERFF human muscle cells and a mouse model of mitochondrial myopathy [51].

Nevertheless, a limitation of the present study is the fact that we did not assess glutamate compartmentalization in the tissue, i.e., whether it is located mainly in the synaptic cleft or in synaptic vesicles, neither proven a higher glutamine deamination into glutamate nor an increased α-ketoglutarate consumption in the Krebs cycle. Therefore, to fully understand the implications of the alterations in amino acid content and distribution in the *Hq* mouse model, more studies are needed.

### Cerebellar Ataxia: The Role of LTD

We also demonstrated, for the first time, lower levels of several proteins involved in glutamate-dependent signaling and neuronal plasticity in the *Hq* mouse cerebellum. Among these alterations, we detected very low levels of the metabotropic glutamate receptor mGluR1. This receptor is expressed in Purkinje cell synapses, outside the active zone, and it is activated when parallel fibers and a climbing fiber excite the Purkinje cell simultaneously, constituting an “error signal” (i.e., of a wrong movement pattern) leading to the weakening of the synapse, which is a primary mechanism of motor learning named LTD [65,66]. The low levels of mGluR1 found in the *Hq* cerebellum suggest an altered motor learning in the surviving neurons, which together with neuronal death would contribute to the development of ataxia. Further supporting this notion, we found significantly lower levels of different proteins acting downstream of mGluR1 in the LTD mechanism, GluRδ2 and HOMER-3, which amplify the intracellular signaling cascade triggered by mGluR1 and play a role in its recycling and regulation, respectively [67,68,69]. Moreover, we observed altered PKCγ activity in the *Hq* cerebellum, which is an important partner for the LTD regulating the Ca^2+^ influx to the postsynaptic density, and the internalization of the AMPA glutamate receptor that results in the synapse weakening [70,71,72]. Thus, in the present work, we also found evidence of disturbances in the final steps of the LTD in the *Hq* mouse. Of note, while changes in most of these proteins were only present at 5–6 months, the phosphorylated form of the GRIA2 subunit from the AMPA receptor was lower in the *Hq* cerebellum from 2 months of age. Prior to this study, an altered AMPA receptor function has been involved in the development of ataxia and epilepsy in the stargazer mouse model lacking stargazin, an essential protein for the maturation, trafficking and correct function of the AMPA receptor [73,74]. In addition, previous studies have related disturbances in mGluR1-mediated signaling and its downstream proteins in genetic and autoimmune human ataxias, as well as in preclinical models of ataxia due to defects in nonmitochondrial genes [75,76,77,78]. Overall, these results show that the ataxia in the *Hq* mouse shares mechanisms of disease with nonmitochondrial ataxias involving an altered LTD mechanism. Although the lower content of these proteins, which are specifically expressed in Purkinje cells could be simply attributed to the Purkinje neuron death previously observed in the cerebellum at these advanced stages of disease [20,29,30,37], the normalization of these proteins by the neuronal markers of Purkinje cells, calbindin, still revealed significant differences between *Hq* and WT mice (Appendix A), ruling out this possibility. Furthermore, some of the observed alterations (i.e., EAAT2 and pGRIA2) were already present at 2 months of age, when granular cell death is still beginning, and Purkinje cell loss is not yet present.

Finally, the astroglial protein AQP4, besides its role in brain water homeostasis, has been reported to modulate synaptic plasticity and LTD, as its knock-down leads to a decreased glutamate uptake and EAAT2 levels, together with an excitotoxicity induction and LTD impairment [79]. Therefore, the higher levels of this protein in the *Hq* cerebellum could act as a compensatory mechanism to counteract the alterations in the glutamate pathway. However, we cannot rule out other putative roles such as an anti-inflammatory effect, as AQP4 deficiency leads to the upregulation of the proinflammatory cytokines TNF-α and IL-1β [80].

Overall, our results suggest novel mechanisms of disease in the *Hq* mouse including hyperammonemia, altered glutamate metabolism and neurotransmission and osmotic regulation, which, with oxidative stress and neuroinflammation, contribute to cerebellar neurodegeneration, probably interacting with each other and creating a vicious cycle of damage that leads to neuronal failure and death. These findings open up the possibility of exploring new therapies based on the attenuation of these disturbances to mitigate the neurological symptoms in MD patients. In this regard, sulforaphane, a compound with antioxidant and anti-inflammatory properties that mitigates excessive GABAergic tone and neuroinflammation in models of hyperammonemic neuropathy [57], could be tested in MD diseases. Nutritional therapies aimed to normalize glutamate metabolism in the central nervous tissue could also be explored. It is worth to mention that our group has demonstrated that supplementation with oral glutamine reduces the abnormally high concentration of glutamate found in cerebrospinal fluid in MELAS patients [43,53], supporting that glutamine treatment could be beneficial in other MD with neurological presentation. Other nutritional approaches such as variations of the classical ketogenic diet, for example high-fat diets, could also attenuate neurological symptoms in MD due to their anti-inflammatory effect as previously reported in the *Hq* mouse model [81]. To specifically treat mitochondrial ataxias, cannabinoid derivatives could also be proposed. Endocannabinoid release from the postsynaptic element is the final step in the LTD mechanism, aiming to minimize the glutamate release from the presynaptic terminal, and limiting neuron overstimulation, glutamate excitotoxicity and neuron death, while promoting the glutamate–glutamine cycle and a balanced glutamate/GABAergic transmission [82]. Therefore, endocannabinoid administration could be a promising therapy to limit excitotoxicity and preserve synaptic plasticity in MD.

In conclusion, the proteomic analysis of the *Hq* mouse cerebellum has revealed previously unreported and relevant disturbances in several neuronal pathways, including glutamate neurotransmission and Ca^2+^ homeostasis, relevant for the LTD mechanism of motor learning. These findings support the interventions aiming to modulate amino acid neurometabolism as a promising therapeutic approach for MD.

## 4. Materials and Methods

### 4.1. Mouse Model

All experimental protocols were approved by an institutional review committee (projects number PROEX 111/15 and PROEX 067/18), authorized and conducted in accordance with European (European Convention for the Protection of Vertebrate Animals ETS123) and Spanish (32/2007 and R.D. 1201/2005) laws on animal protection in research and with Animal Research: Reporting of in vivo Experiments (ARRIVE) guidelines. For the present study, heterozygous “*Harlequin*” mice (*Hq*/Y, hereafter named *Hq*, B6CBACa A*w*-*J*/A-Aifm*1*Hq/J), heterozygous females (X/*Hq*, B6CBACa Aw-J/A-Aifm1Hq/J) and “wild-type” (WT, B6CBACa) male mice of the same strain from The Jackson Laboratory (Bar Harbor, ME, USA) were used. Six-to-eight-week-old heterozygous females (X/*Hq*, B6CBACa Aw-J/A-Aifm1Hq/J) were crossed with wild-type males (WT, B6CBACa) for breeding and maintenance of the colony. F2 generation mice were used for the experimental procedures. Mice were housed at the animal facility of Hospital 12 de Octubre in controlled conditions of temperature, humidity and ventilation, with 12 h light/dark cycles and ad libitum access to food and water. Mouse genotyping was performed according to standardized protocols. DNA was extracted from the mice’s tails, and primers were designed as previously reported by [20] and purchased from Integrated DNA Technologies (Coralville, IA, USA).

### 4.2. Study Design

#### 4.2.1. Proteomic Study

To analyze the impact of the disease in the cerebellar proteome of the *Hq* mouse and to identify the main cellular pathways affected by the mitochondrial defect, 6–8-week-old male mice (23 WT and 21 *Hq* mice, project number PROEX 111/15) were obtained from The Jackson Laboratory and maintained at the animal facility of Hospital 12 de Octubre in controlled conditions of temperature, humidity and ventilation, with 12 h light/dark cycles and ad libitum access to food and water. Eight weeks after the onset of clear symptoms of ataxia in the *Hq* mice and at the age of 5.2 ± 0.2 and 5.3 ± 0.1 months for WT and *Hq*, respectively, mice were sacrificed by cervical dislocation, and the brain and cerebellum were quickly dissected and frozen in liquid nitrogen before storage at −80 °C until the proteomic analysis and validation [29].

#### 4.2.2. Neurodegeneration Progression Study

To assess the disease progression on the cellular pathways affected by the mitochondrial defect, 8-week-old heterozygous females (X/*Hq*, B6CBACa Aw-J/A-Aifm1Hq/J) were crossed with WT males of the same strain (all purchased from The Jackson Laboratory), and the tissues of the male mice from the F1 generation were used for the analysis at different ages. PCR genotyping of F1 mice was performed on tail DNA using previously described primers [20].

F1 male mice were sacrificed at 2, 3 and 6 months (8 WT and 8 *Hq* mice for each age, project number PROEX 067/18) by cervical dislocation, and the brain and cerebellum were dissected, frozen in liquid nitrogen and stored at −80 °C for the subsequent biochemical analysis.

### 4.3. Tissue Processing

The right hemisphere (midsagittal section) was fixed in 4% paraformaldehyde and paraffin-embedded in blocks for histology (5 µm sections). The left cerebellum was immediately snap-frozen in liquid nitrogen and stored at −80 °C for molecular analysis. Cerebellar tissue homogenates were prepared in ice-cold RIPA buffer (50 mM Tris-HCl pH 7.4, 1% NP-40, 0.5% Na-deoxycholate, 1% SDS, 150 mM NaCl, 2 mM EDTA) plus protease and phosphatase inhibitors (Roche Diagnostics Corp, Indianapolis, IN, USA) with a Potter homogenizer. Protein concentration was determined with the BCA Assay Kit (Pierce, Thermo Scientific, Waltham, MA, USA).

### 4.4. Western Blotting

Samples of total tissue homogenates were loaded onto SDS-PAGE gels (7.5 to 15%). Resolved proteins were transferred to PVDF membranes, blocked with 5% skimmed milk or bovine serum albumin, incubated with primary (see Appendix A) and horseradish peroxidase conjugated secondary antibodies and finally developed with the ECL Prime Western blotting detection reagent (Amersham GE Healthcare, Little Chalfont, UK). Data were normalized using ƴ-tubulin as loading control. A densitometry analysis was performed with ImageJ 1.8 software (Rasband, W.S, ImageJ; U.S. National Institutes of Health, Bethesda, MA, USA).

### 4.5. Immunofluorescence

Immunofluorescence in paraffin-embedded tissues slides was performed for multiple-protein staining. After deparaffinization and rehydration, antigen retrieval was performed in citrate buffer pH 6.0 or Tris-EDTA buffer pH 9.0, at 95–100 °C for 20 min. Slides were permeabilized and blocked in 2% BSA, 0.1% Triton X-100 in Tris-buffered saline (TBS) solution, incubated with the corresponding primary antibodies (Appendix A), washed and incubated with fluorescent secondary antibodies. Slides were finally incubated with 4′,6-diamidino-2-phenylindole (DAPI) and mounted with Dako Fluorescent Mounting Medium (Agilent Technologies Inc., Santa Clara, CA, USA). The visualization of slides and image processing were performed with a Zeiss LSM510 META confocal scanning microscope (Carl Zeiss MicroImaging GmbH, Jena, Germany).

### 4.6. Proteomic Analysis

A differential proteomic profile study was performed in cerebellum homogenates from WT and *Hq* mice at the Proteomic Service of Severo Ochoa Molecular Biology Center “CBMSO” (Madrid, Spain). A pool of cerebellar proteins with 150 μg of total protein was obtained by mixing an equal amount of protein homogenate from each animal of each experimental group. The whole proteome was concentrated in the stacking/resolving gel interface, and the unseparated protein bands were visualized by Coomassie staining and excised. Gel pieces were destained in acetonitrile:water 1:1, reduced with 10 mM DTT, and thiol groups were alkylated with 10 mM 1,4-dithiothreitol for 1 h at 56 °C and subsequently with 50 mM iodoacetamide during 1 h at room temperature in the dark. Next, proteins were digested in situ with sequencing-grade trypsin (Promega, Madison, WI, USA), dried down, desalted and resuspended in 0.1% formic acid, then analyzed by RP-LC-MS/MS with nano HPLC Easy-nLC II equipment coupled to an LTQ-Orbitrap-Velos mass spectrometer (Thermo Scientific, Waltham, MA, USA). Resolved peptides were identified by a database search in UniProt-Mus musculus, using the Sequest algorithm as a search engine and Proteome Discoverer 1.4 software (Thermo Scientific, Waltham, MA, USA). Relative quantification was performed by the iTRAQ procedure. Each sample was marked with two different iTRAQ reagents. This technique allows peptides from each sample to be differentially marked and identified. The signal intensity for each fragment is normalized by the value found for that peptide in the WT group, which allows the relative quantification between experimental groups. The peptide mixture from the proteins’ tryptic digest (50µg) of each experimental group was labeled using the iTRAQ reagent multiplex kit (reagents 113, 115, 117 and 119) (Applied Biosystems, Waltham, MA, USA). Briefly, peptides were dissolved in 0.5 M triethylammonium bicarbonate and adjusted to pH 8.0. For labeling, each iTRAQ reagent was dissolved in isopropanol and added to the respective peptide mixture, which was then incubated at room temperature for two hours. Labelling was stopped by the addition of 0.1% formic acid. Whole SN were dried down and the four samples were mixed to obtain the multiplex-labeled mixture. The mixture was desalted onto OMIX pipette tips C18 (Agilent Technologies, Santa Clara, CA, USA) for the mass spectrometric analysis, according to previously described methods [83].

### 4.7. Gene Expression Assay

For the gene expression analysis in the cerebellum, the Quantigene™ Plex Assay was run (Thermo Fisher Scientific; Waltham, MA, USA). First of all, RNA was isolated from 5 mg of tissue in commercial homogenization buffer (Invitrogen, Carlsbad, CA, USA) following the manufacturer’s instructions. After extraction, the RNA from samples was hybridized in a plate with the detection probe at 54 °C for 16 h and a gentle agitation at 400 rpm. Next, the signal corresponding to the mRNA of interest was amplified through successive hybridization with preamplifier, amplifier and label probes and subsequently quantified with the addition of a chemiluminescent substrate. The signal was measured on a Luminex MAGPIX instrument (EMD Millipore Corporation, Billerica, MA, USA) and the data of each mRNA expression were normalized to the geometric mean of three genes selected as internal controls (*Tfrc*, *Hprt* and *Rpl32*).

### 4.8. High-Performance Liquid Chromatography

The analysis of high-performance liquid chromatography (HPLC) of brain and cerebellar homogenates was carried out using a methodology approved by the National Accreditation Entity ENAC (ISO15189) and used for the processing of clinical samples in the Service of Biochemistry of the Hospital 12 de Octubre. Briefly, 200 µL of total homogenates containing 18.2 mg of total wet tissue (protein concentration 10–12 µg/µL) was deproteinized with 50% sulfosalicylic acid and analyzed by ion exchange chromatography with postcolumn derivatization with ninhydrin in the Biochrom 30+ equipment (Biochrom Ltd., Cambridge, UK). Amino acid peaks were detected and quantified with OpenLAB EZChrom Edition software A.04.10 (Siegwerk Druckfarben AG & Co. KGaA; Siegburg, Germany), and the mean concentration for each metabolite was expressed as μM.

### 4.9. Statistical Analysis

All study variables are presented as mean ± standard deviation. The nonparametric Kruskal–Wallis test was used to assess whether a significant “group” effect was found for the different variables, in which case pairwise comparisons were performed post hoc with Tukey’s test and Mann–Whitney’s U test for comparisons between two groups (WT vs. *Hq*). Statistical significance was set at *p* value < 0.05. All statistical analyses were performed with GraphPad Prism^®^ 6 for Windows (GraphPad Software, San Diego, CA, USA).

## Figures and Tables

**Figure 1 ijms-24-10973-f001:**
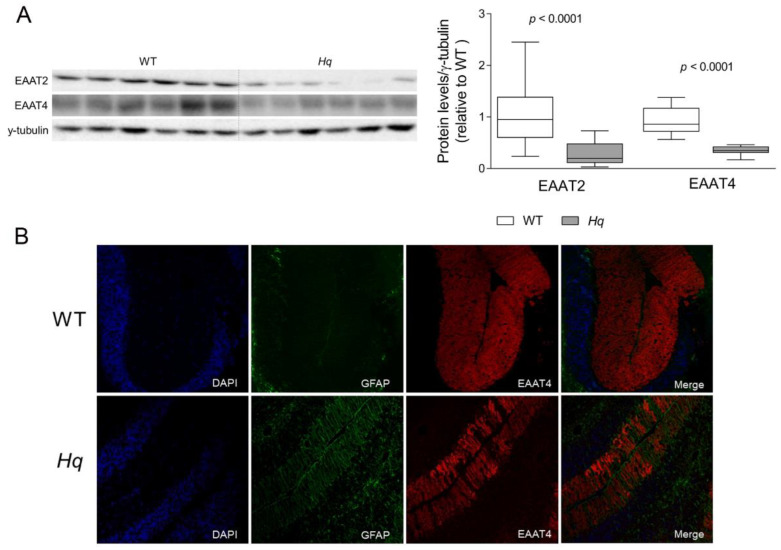
Glutamate transporters. (**A**) Representative Western blot and densitometry analysis of glutamate transporters EAAT2 and EAAT4 in the cerebellum of WT and *Hq* mice. γ-tubulin was used as loading control. Data (mean, interquartile range and min and max values) are expressed relative to the WT group. *p*-Values for differences between WT and *Hq* groups (Mann–Whitney U test) are shown above the graph. (**B**) Representative immunofluorescence of GFAP (green), EAAT4 (red) and nuclei staining with DAPI (blue) in cerebellar tissue of WT and *Hq* mouse. Abbreviations for groups: WT, wild type; *Hq*, *Harlequin*.

**Figure 2 ijms-24-10973-f002:**
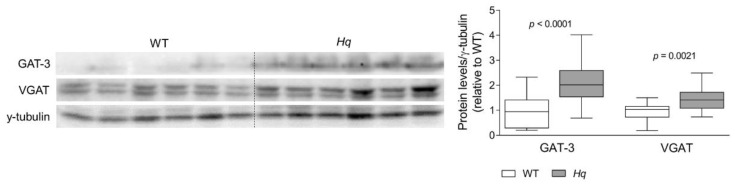
GABA transporters. Representative Western blot and densitometry analysis of GABA transporters GAT-3 and VGAT in the cerebellum of WT and *Hq* mice. γ-tubulin was used as loading control. Data (mean, interquartile range and min and max values) are expressed relative to the WT group. *p*-Values for differences between WT and *Hq* groups (Mann–Whitney U test) are shown above the graph. Abbreviations for groups: WT, wild type; *Hq*, *Harlequin*.

**Figure 3 ijms-24-10973-f003:**
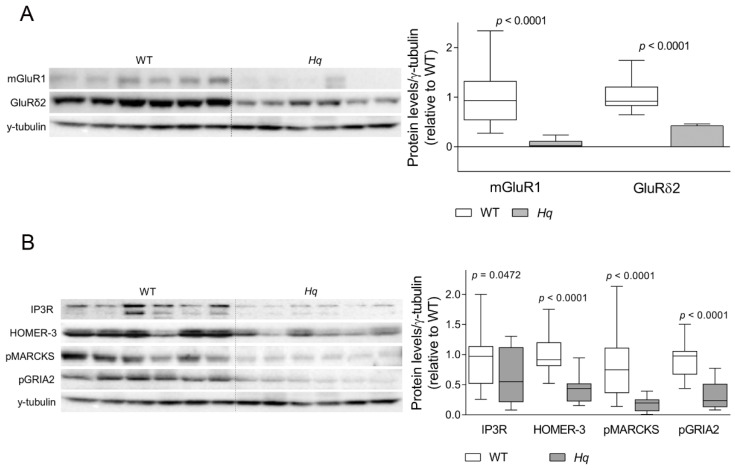
Glutamate receptors and Ca^2+^ homeostasis-related proteins. Representative Western blot and densitometry analysis of (**A**) glutamate receptors mGluR1 and GluRδ2 and (**B**) Ca^2+^ homeostasis-related proteins, Homer protein homolog 3 (HOMER-3), inositol 1,4,5-triphosphate receptor type 1 (IP_3_R), myristoylated alanine-rich C-kinase substrate phosphorylated at Serine 152/156 (pMARCKS) and subunit 2 of the glutamate ionotropic receptor AMPA type phosphorylated at Serine 880 (pGRIA2) in the cerebellum of WT and *Hq* mice. γ-tubulin was used as loading control. Data (mean, interquartile range and min and max values) are expressed relative to the WT group. *p*-Values for differences between WT and *Hq* groups (Mann–Whitney U test) are shown above the graph. Abbreviations for groups: WT, wild type; *Hq*, *Harlequin*.

**Figure 4 ijms-24-10973-f004:**
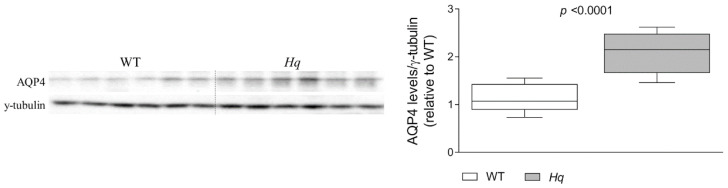
Aquaporin 4. Representative Western blot and densitometry analysis of aquaporin 4 in the cerebellum of WT and *Hq* mice. γ-tubulin was used as loading control. Data (mean, interquartile range and min and max values) are expressed relative to the WT group. Mann–Whitney U test, significantly different compared to WT. Abbreviations for groups: WT, wild type; *Hq*, *Harlequin*.

**Table 1 ijms-24-10973-t001:** Proteins showing differential expression between *Harlequin* and wild-type mice.

Accession	Description	Gene	Score	Coverage	Proteins	Unique Peptides	Peptides	WT	*Hq*
**Glial proteins**									
P03995	Glial fibrillary acidic protein (GFAP)	*Gfap*	37.84	26.05	1	10	11	1	1.873
A0A0A6YWC8	Vimentin (VIM)	*Vim*	12.21	15.69	4	5	7	1	1.713
**Neurotransmission**									
P31650	Sodium- and chloride-dependent GABA transporter 3 (GAT-3)	*Slc6a11*	24.81	9.89	1	5	5	1	1.176
O35633	Vesicular inhibitory amino acid transporter (VGAT)	*Slc32a1*	14.03	6.29	1	3	4	1	1.181
O35544	Excitatory amino acid transporter 4 (EAAT4)	*Slc1a6*	5.31	4.63	1	1	2	1	0.692
E9Q517	Sodium- and chloride-dependent glycine transporter 1(Gly-T1)	*Slc6a9*	8.02	5.37	3	3	3	1	0.953
**Cellular signaling and Ca^2+^ handling**									
F6RT34	Myelin basic protein (MBP)	*Mbp*	79.69	38.68	7	9	9	1	1.213
A0A087WSI9	Synaptosomal-associated protein 91 (SNAP91)	*Snap91*	2.82	9.13	6	2	2	1	1.292
Q8R5L1	Complement component 1 Q subcomponent-binding protein, mitochondrial C1qbp (C1QBP)	*C1qbp*	2.30	11.11	1	2	2	1	1.172
P11881	Inositol 1,4,5-trisphosphate receptor type 1 (IP3R1)	*Itpr1*	86.55	10.95	3	29	29	1	0.856
Q99JP6	Homer protein homolog 3 (HOMER-3)	*Homer3*	7.99	6.74	2	3	3	1	0.808
Q2NKI4	Protein kinase C (PKC)	*Prkcg*	6.12	3.10	2	2	2	1	0.726
P62748	Hippocalcin-like protein 1 (HCAL1)	*Hpcal1*	16.15	19.69	4	4	4	1	0.800
**Osmotic regulation**									
P07724	Serum albumin (ALB)	*Alb*	83.51	25.49	1	14	14	1	1.313
P55088	Aquaporin-4 (AQP4)	*Aqp4*	17.79	11.15	1	3	3	1	1.187
**OXPHOS components**									
Q91VD9	NADH-ubiquinone oxidoreductase 75 kDa subunit (NDUFS1)	*Ndufs1*	22.70	17.19	2	10	10	1	0.811
D3YXT0	NADH dehydrogenase [ubiquinone] iron-sulfur protein 2 (NDUFS2)	*Ndufs2*	8.39	6.64	4	3	3	1	0.791
Q9DCT2	NADH dehydrogenase [ubiquinone] iron-sulfur protein 3 (NDUFS3)	*Ndufs3*	20.23	24.71	1	5	5	1	0.739
Q9DC69	NADH dehydrogenase [ubiquinone] 1 alpha subcomplex subunit 9, mitochondrial (NDUFA9)	*Ndufa9*	13.82	14.06	1	5	5	1	0.729
Q99LC3	NADH dehydrogenase [ubiquinone] 1 alpha subcomplex subunit 10, mitochondrial (NDUFA10)	*Ndufa10*	6.41	14.65	1	4	4	1	0.681
Q62425	Cytochrome c oxidase subunit NDUFA4 (NDUFA4)	*Ndufa4*	5.37	26.83	1	2	2	1	0.732
P12787	Cytochrome c oxidase subunit 5A, mitochondrial (COX5A)	*Cox5a*	8.37	15.07	1	3	3	1	0.673
**Miscellaneous**									
A8DUK4	Beta-globin (β-globin)	*Hbbt1*	142.86	78.91	2	3	9	1	1.277
Q91VB8	Alpha globin 1 (α-globin 1)	*Hba-a2*	57.68	52.11	2	6	6	1	1.268
Q00623	Apolipoprotein A-I (APOA-1)	*ApoA-1*	11.06	15.15	1	4	4	1	1.225
O08709	Peroxiredoxin-6 (PRDX6)	*Prdx6*	32.68	39.29	4	8	8	1	1.209
Q9Z1N5	Spliceosome RNA helicase Ddx39b (DDX39B)	*Ddx39b*	2.76	5.61	1	2	2	1	0.808

Description and database ID at https://www.uniprot.org/, (accessed on 29 September 2016) mouse protein name (abbreviated name), gene name, score, coverage, proteins, unique peptides, peptides and mean values of the relative levels of each protein marked with two independent fluorophores and expressed relative to the value obtained for the protein in the wild-type sedentary group stained with the first fluorophore are shown for each experimental group. Abbreviations: WT, wild type; *Hq, Harlequin*.

**Table 2 ijms-24-10973-t002:** Time course study of cerebellum variables.

		2 Months			3 Months			6 Months		
	N with Data	WT	*Hq*	*p*-Value	WT	*Hq*	*p*-Value	WT	*Hq*	*p*-Value
EAAT2	8	100 ± 23	91 ± 28	0.898	102 ± 29	69 ± 14	0.073	88 ± 15	58 ± 24	0.072
EAAT4	8	100 ± 30	73 ± 36	**0.038**	69 ± 29	35 ± 19	**0.007**	80 ± 50	9 ± 6	**0.015**
GAT-3	8	100 ± 69	91 ± 101	0.932	46 ± 32	80 ± 42	0.129	49 ± 55	107 ± 39	**0.028**
VGAT	8	100 ± 64	99 ± 47	0.854	100 ± 49	146 ± 54	0.104	105 ± 43	172 ± 68	**0.038**
mGluR1	8	100 ± 35	96 ± 42	0.854	99 ± 38	117 ± 53	0.495	77 ± 21	33 ± 22	**0.0011**
GluRδ2	8	100 ± 23	103 ± 54	0.932	56 ± 22	38 ± 19	0.130	79 ± 65	7 ± 3	**0.003**
HOMER-3	8	100 ± 36	121 ± 34	0.193	149 ± 54	104 ± 48	0.065	124 ± 77	5 ± 4	**0.01**
IP3R1	8	100 ± 50	199 ± 39	0.083	142 ± 58	117 ± 49	0.560	142 ± 104	8 ± 4	**0.038**
pMARCKS	8	100 ± 64	109 ± 36	0.494	82 ± 28	84 ± 24	0.494	62 ± 43	11 ± 9	**0.015**
pGRIA2	8	100 ± 23	67 ± 32	**0.026**	91 ± 22	70 ± 15	**0.05**	104 ± 18	59 ± 23	**0.0011**
NeuN	8	100 ± 23	105 ± 23	**0.433**	117 ± 36	88 ± 23	**0.159**	109 ± 58	27 ± 14	**0.0281**
Calbindin	8	100 ± 32	129 ± 37	**0.375**	118± 23	108 ± 40	**0.932**	101 ± 55	28 ± 14	**0.0379**

Time course assessment of the cerebellar proteins showing altered expression in the *Hq* mice in the proteomic study. Variables are shown for WT and *Hq* mice at 2, 3 and 6 months. Data are shown as mean ± SD and expressed as percentage of the 2-month-old WT group. Significant *p*-values for Mann–Whitney U test for WT vs. *Hq* are shown in bold. Abbreviations: WT, wild type; *Hq*, *Harlequin*.

**Table 3 ijms-24-10973-t003:** Amino acids and amino acid-derived metabolites in cerebellum and brain of 6- and 2-month-old mice.

	6 Months				6 Months				2 Months			
	Cerebellum				Brain				Cerebellum			
Metabolite	N	WT	*Hq*	*p*-Value	N	WT	*Hq*	*p*-Value	N	WT	*Hq*	*p*-Value
Alanine	5	14.0 ± 0.8	15.1 ± 1.7	0.532	5	20.2 ± 1.7	29.9 ± 4.2	**0.0079**	5	24.0 ± 2.1	42.1 ± 25.1	**0.0286**
Ammonium	5	68.6 ± 6.1	84.6 ± 9.7	**0.0079**	5	61.8 ± 7.0	62.7 ± 4.1	0.802	5	166.2 ± 27.2	198.4 ± 49.9	0.532
Arginine	5	5.7 ± 0.9	6.4 ± 1.1	0.413	5	5.9 ± 1.0	5.5 ± 1.2	0.310	5	11.6 ± 2.3	14.2 ± 5.0	0.532
Cystathionine	5	40.8 ± 0.2	41.3 ± 0.8	0.413	5	9.8 ± 0.3	10.4 ± 2.2	0.667	5	79.5 ± 9.9	85.8 ± 12.9	0.310
GABA	5	43.6 ± 8.4	60.5 ± 14.3	**0.0317**	5	72.1 ± 11.4	76.4 ± 10.7	0.802	5	118.9 ± 28.5	150.0 ± 33.7	0.222
Glutamine	5	106.5 ± 19.3	103.2 ± 13.6	0.532	5	102.3 ± 4.3	124.4 ± 10.5	**0.0317**	5	212.1 ± 32.4	306.6 ± 25.7	**0.0079**
Glutamate	5	203.8 ± 36.2	119.9 ± 17.8	**0.0079**	5	234.8 ± 12.7	216.9 ± 20.6	0.095	5	466.6 ± 70.4	459.5 ± 53.6	0.944
Histidine	5	1.6 ± 0.2	1.6 ± 0.3	0.683	5	1.6 ± 0.1	1.5 ± 0.2	0.413	5	3.7 ± 0.7	4.7 ± 0.8	0.111
Homocysteine	5	2.2 ± 0.2	2.1 ± 0.2	0.413	5	0.8 ± 0.4	0.6 ± 0.4	0.057	5	9.7 ± 2.1	9.2 ± 2.1	0.667
Leucine	5	10.0 ± 2.0	12.2 ± 1.2	0.0952	5	3.6 ± 0.3	4.2 ± 0.5	0.056	5	11.7 ± 4.2	16.0 ± 3.5	0.151
Lysine	5	3.9 ± 1.0	6.1 ± 0.8	0.0556	5	4.5 ± 0.6	3.7 ± 0.5	0.056	5	11.2 ± 1.4	15.6 ± 3.5	0.056
Phenylalanine	5	1.4 ± 0.3	1.1 ± 0.8	0.999	5	1.1 ± 0.2	1.4 ± 0.2	0.064	5	4.0 ± 0.5	4.8 ± 1.0	0.286
Phosphoserine	5	5.6 ± 1.3	4.4 ± 0.5	0.151	5	3.1 ± 0.4	3.5 ± 0.7	0.151	5	16.7 ± 7.3	17.7 ± 3.3	0.151
Serine	5	9.9 ± 1.2	7.2 ± 0.4	**0.0079**	5	16.8 ± 2.0	16.3 ± 1.4	0.532	5	24.7 ± 4.3	20.3 ± 3.7	0.222
Taurine	5	173.8 ± 50.6	162.7 ± 26.0	0.413	5	220.2 ± 29.7	244.1 ± 33.2	0.310	5	336.8 ± 68.8	402.6 ± 54.1	0.222
Threonine	5	6.6 ± 1.4	7.1 ± 1.4	0.413	5	6.7 ± 0.8	8.0 ± 1.4	0.151	5	16.7 ± 2.1	25.9 ± 5.2	**0.0079**
Tyrosine	5	1.8 ± 0.3	1.2 ± 0.5	0.508	5	1.6 ± 0.2	1.5 ± 0.2	0.532	5	4.0 ± 1.0	5.1 ± 1.1	0.151
Urea	5	77.7 ± 11.9	94.8 ± 16.2	0.151	5	90.2 ± 6.4	117.2 ± 18.8	0.056	5	227.5 ± 21.8	207.9 ± 22.8	0.310

Data are shown as mean ± SD (μmoL). Significant *p*-values for Mann–Whitney U test for WT vs. *Hq* are shown in bold. Abbreviations: GABA, γ-aminobutiric acid; N, number of data points in both WT and *Hq* experimental groups; WT, wild type; *Hq, Harlequin.*

## Data Availability

Data are available as Appendix A.

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
