# Peer review of "Pathophysiology of Cerebellar Degeneration in Mitochondrial Disorders: Insights from the Harlequin Mouse"

_ijms, 2023, doi:10.3390/ijms241310973_

Round 1

Reviewer 1 Report

The Manuscript: „Pathophysiology of cerebellar degeneration in mitochondrial disorders: Insights from the Harlequin mouse.’’ by Miguel Fernández de la Torre and colleagues analyses the cellular pathways implicated in the neuronal death in the Harlequin mouse model of mitochondrial disorders through an elaborated proteomic approach. The manuscript is nicely written with a good description of methods and impressive results. After going through the manuscript, I have following comments for the authors:

1.     Previous studies have shown that the key insight of studies on Hq Mouse is the role of oxidative stress in cerebellar degeneration. Mitochondrial dysfunction is known to lead to an imbalance between the production of reactive oxygen species (ROS) and the cell's ability to detoxify them. Was this aspect studied in the current study?

2.     The study gives a valuable insights into the pathophysiology of cerebellar degeneration in mitochondrial disorders. Understanding various underlying mechanisms such as - role of oxidative stress, impaired mitochondrial bioenergetics, calcium dysregulation, and inflammation - is crucial for the development of targeted therapies to mitigate cerebellar dysfunction and improve the quality of life for individuals with mitochondrial disorders. The secondary aim of the study is also to identify potential therapeutic targets to attenuate neurodegeneration. However, this issue is not discussed in detail. Please elaborate these points in the discussion section.

3.     The sequential order of the sections and sub sections of the manuscript is bit awkward. I would suggest moving up the material and methods after the introduction and before results.

English is fine. Some grammatical corrections and syntax adjustment are needed. 

Reviewer 2 Report

The paper describes proteomic and metabolic changes in the cerebellum of the Hq mouse in response to neurodegeneration due to AIF deficiency.

1. The main problem is the missing link of the proteomic data to the potential alteration in brain tissue composition due to ongoing cell death. That should be provided by standard means of determining neuronal cell counts at the different time points.

2. The investigated proteins should be named in table 1 as mentioned in the text.

3. The observed metabolite changes in the tissue homogenates (table 3) are adding very little information, since they are difficult to interprete due to compartmentalisation issues (e.g. glutamate is located in synaptic vesicles). Instead immunolabelled tissue sections would be much more informative.

4. In conclusion, the authors should try to explain if their data are a result of a tissue composition change or a change of protein expression levels of neurons or glial cells.

None.

Round 2

Reviewer 2 Report

The authors have addressed all of my comments appropriately.

None.